# Fairness and Efficiency in Online Class Matching

**MohammadTaghi Hajiaghayi**
University of Maryland

**Shayan Chashm Jahan**
University of Maryland

**Mohammad Sharifi**
Sharif University of Technology

**Suho Shin**
University of Maryland

**Max Springer**
University of Maryland

## Abstract

The online bipartite matching problem, extensively studied in the literature, deals with the allocation of online arriving vertices (items) to a predetermined set of offline vertices (agents). However, little attention has been given to the concept of class fairness, where agents are categorized into different classes, and the matching algorithm must ensure equitable distribution across these classes.

We here focus on randomized algorithms for the fair matching of indivisible items, subject to various definitions of fairness. Our main contribution is the first (randomized) non-wasteful algorithm that simultaneously achieves a $1/2$ approximation to class envy-freeness (CEF) while simultaneously ensuring an equivalent approximation to the class proportionality (CPROP) and utilitarian social welfare (USW) objectives. We supplement this result by demonstrating that no non-wasteful algorithm can achieve an $\alpha$-CEF guarantee for $\alpha > 0.761$. In a similar vein, we provide a novel input instance for deterministic divisible matching that demonstrates a nearly tight CEF approximation.

Lastly, we define the "price of fairness," which represents the trade-off between optimal and fair matching. We demonstrate that increasing the level of fairness in the approximation of the solution leads to a decrease in the objective of maximizing USW, following an inverse proportionality relationship.

## 1 Introduction

The rapid advancement of technology and the widespread adoption of online platforms have revolutionized the way we interact, conduct business, and access services. From ride-sharing platforms [5] to online marketplaces [32], these platforms connect users with a vast array of resources, creating unprecedented opportunities for dynamic resource allocation. However, efficiently matching supply with demand in such *online* environments poses significant challenges, necessitating the exploration of novel algorithms and strategies.

The fundamental online matching problem lies at the core of resource allocation in such platforms. Unlike traditional matching problems where the entire set of agents and resources are known in advance, the online matching problem involves making real-time decisions without complete information about future arrivals and requests. This inherent uncertainty and dynamic nature render traditional static matching algorithms inadequate, demanding the development of new techniques tailored specifically for online settings.

The objective of the online matching problem is to match as many arriving goods to static (offline) agents as possible in an efficient manner. The realization of this objective may vary depending on the specific application context. However, regardless of the objective, the challenge lies in making immediate decisions while accounting for future arrivals and the scarcity of resources. The performance evaluation of algorithms designed for this problem is based on their competitive ratio,

38th Conference on Neural Information Processing Systems (NeurIPS 2024).

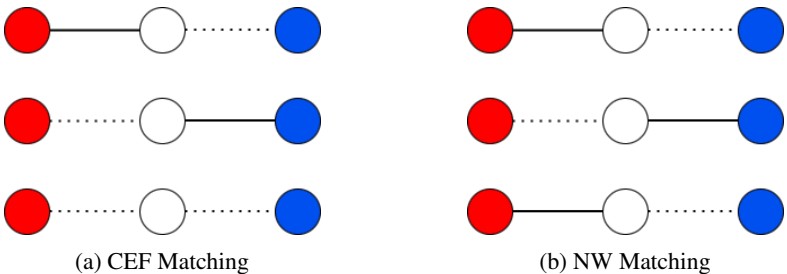

(a) CEF Matching          (b) NW Matching

Figure 1: Examples of class envy-free (CEF) and non-wasteful (NW) matchings where bolded lines indicate a matching. Red nodes indicate agents in the first class, blue nodes indicate agents in the second class, and white nodes indicate items.

representing the worst-case approximation ratio between the size of the produced matching and the maximum possible size with complete hindsight. It is well known that the best deterministic algorithm can only achieve a $1/2$-approximation and the seminal work of Karp et al.[28] provides a randomized algorithm with a tight $(1 - 1/e)$-approximation guarantee.

To motivate the study of online matching under fairness constraints [21], we turn to the real-world challenges presented by modern Internet economics and emerging marketplaces. These settings demand solutions that balance transparency with fairness, as emphasized in Moulin's "Fair Division in the Internet Age" [35]. In various applications, such as allocating advertisement slots [32], assigning packets in switch routing [4], distributing food donations [34], and matching riders to drivers in ridesharing platforms [13], items or services must be matched to agents immediately and irrevocably as they arrive. However, much of the existing work overlooks the need for fairness in matching decisions. Consider, for instance, a food bank that must allocate perishable food items upon arrival. Ensuring that these resources are distributed equitably across all communities is crucial to addressing fairness concerns in these high-stakes scenarios.

It is for precisely this reason that [21] initiate the study of *class fair matchings* where a set of items arriving online must be assigned to agents, who are partitioned into known classes, with the goal of achieving fairness among classes. In this problem, similar to online bipartite matching, agents either like an item (value 1) or do not like it (value 0), however our objective is to ensure equitable treatment of different classes. We refer to the standard unfair objective that maximizes the number of matched agents as the utilitarian social welfare (USW). When considering classes, the notion of class envy-freeness (CEF) ensures that no class of agents can enhance their overall value by obtaining the items allocated to another class, even if the items are optimally distributed within their own class. It is important to note that in cases involving indivisible items, a class envy-free matching may not always be possible (see Figure 1). Consequently, our research mainly focuses on addressing the central *unresolved question* in the online class fair matching problem posed by [21]:

*Open Problem* 1. Can a randomized algorithm for matching indivisible items achieve any reasonable CEF approximation together with either non-wastefulness or a USW approximation?

## 1.1 Our Results

Our work mainly focuses on *randomized* algorithms for matching *indivisible* items in a fair manner, subject to the varying definitions of fairness adapted from the fair division literature. Notably, we provide the first non-wasteful algorithm that simultaneously obtains approximate class fairness guarantees in expectation, resolving Open Problem 1 posed by [21] in an affirmative manner. This algorithm is a natural random matching procedure that exhibits notable constant factor approximations in spite of its simplicity. Our main algorithmic result is stated formally as follows.

**Theorem 1.1** (Randomized algorithm; informal). *For randomized matching of indivisible items, the* RANDOM *algorithm satisfies non-wastefulness, $\frac{1}{2}$-CEF, $\frac{1}{2}$-CPROP, and $\frac{1}{2}$-USW.*

We highlight that the analysis of the various approximate fairness guarantees is highly non-trivial due to the non-additive nature of the objective functions, to be discussed more formally in Section 3. In exploring the tightness of our algorithmic guarantees, we additionally construct an upper bound adversarial input that demonstrates the limits on achievable fairness in this problem setting.

| **Indivis.** | USW | CEF | CPROP | **Divis.** | USW | CEF | CPROP |
|---|---|---|---|---|---|---|---|
| Alg. | $1/2$ | $1/2$ | $1/2$ | Alg. | $1/2$ [21] | $1-1/e$ [21] | $1-1/e$ [21] |
| Bound | $1-1/e$ [28] | $\frac{e^2-1}{e^2+1}$ | $1-1/e$ [1] | Bound | $1-1/e$ [28] | $0.67$ | $1-1/e$ [21] |

Table 1: The summary of our results on randomized algorithms. Each algorithm achieves its three guarantees simultaneously, while the upper bound holds for any algorithm, separately for each guarantee. Results from prior works in the divisible setting are noted with citation for completeness.

**Theorem 1.2** (Indivisible CEF upper bound; Informal). *Any non-wasteful (possibly randomized) algorithm cannot achieve an $\alpha$-CEF guarantee for $\alpha > \frac{e^2-1}{e^2+1} \approx 0.761$.*

This upper bound construction builds upon the results of [28] to demonstrate that CEF cannot be achieved and will be presented in Section 4.1 with a complete analysis found in Section B.3. We additionally note that while our results show a deviation in the guarantees from traditional fair division problems, certain properties nicely translate to our problem setting – we expound on this fact in Appendix A with a comprehensive discussion on the connections between *class* Nash welfare and CEF1.

In Section 4.2, we additionally provide a strengthened upper bound for the *divisible* matching setting through a careful input construction which further resolves an open problem left by [21].

**Theorem 1.3** (Divisible CEF upper bound; Informal). *No deterministic algorithm for divisible matching can achieve $\beta$-CEF for any $\beta > 0.677$.*

Finally, in an effort to further quantify the inherent gap between fair and unfair solutions to the online matching problem, we conclude the paper with a definition for the "price of fairness" which is intuitively the necessary trade-off between an optimal and a fair matching. We demonstrate that as we strive for a higher level of fairness in the approximation of the solution, the objective of maximizing utilitarian social welfare (USW) deteriorates, exhibiting an inverse proportionality relationship.

**Theorem 1.4** (Price of fairness; Informal). *For any $\varepsilon > 0$, there exists a problem instance such that no (possibly randomized) online algorithm that guarantees $\alpha$-CEF can achieve USW larger than $\frac{1}{1+\alpha} + \varepsilon$.*

## 1.2 Related Work

**Online Matching.** For an extensive exploration of the vast literature on online matching, we refer readers to [33], while summarizing key findings relevant to this paper. The seminal work by [28] introduces the RANKING algorithm which, in our problem setting, corresponds to a randomized algorithm for matching indivisible items that achieves a utilitarian social welfare (USW) approximation of $1 - 1/e$. In the fractional matching domain, an identical result is achieved with a deterministic algorithm [26]. The literature further explores randomized input models of online matching problems to surpass this well-known $1 - 1/e$ barrier. For online vertices that arrive in a random order (reducing the power of an adversarial input) [31] and [27] demonstrate that the competitive ratio of the Ranking algorithm falls between 0.696 and 0.727. Moreover, [24] propose a variant of Ranking that surpasses the $1 - 1/e$ barrier in vertex-weighted online matching under random-order arrivals, with a further improvement to 0.668 presented by [25]. In stochastic matching, where items are drawn from a known distribution, the best-known competitive ratios for unweighted and vertex-weighted online stochastic matching are 0.711 and 0.700, respectively [15, 23].

**Fair Division.** In the offline setting, envy-freeness (EF) and proportionality (PROP), along with their approximations, are commonly employed as criterions of fairness in the allocation of items to agents. For divisible items, an allocation which is envy-free and pareto optimal (PO) always exists [38] and can be computed efficiently when agent valuation functions are additive [14]. For indivisible items, such allocations are not guaranteed to exist. As such, two relaxations are commonly studied: envy-freeness up to one item (EF1) [30] and maximin share fairness (MMS) [10]. An allocation that satisfies EF1 is guaranteed to exist when valuations are monotone, and are further PO when agents have additive valuations [12]. However, the existence of MMS allocations is not guaranteed, even

---

[1]This fact trivially holds when considering the upper-triangular construction of [28] and $k = 1$ in our setting.

for additive valuations. Nevertheless, there are various polynomial time approximation algorithms [17, 18, 22, 20, 36]. In our fair matching problem, we effectively compute an online allocation that satisfies the aforementioned fairness criteria by treating each class as an agent within the system.

**Fair Online Matching.** Most closely related to our work is the preliminary work of [21] that introduces the online class fair problem. This paper provides the initial results for approximate guarantees on the objectives of class envy-freeness (CEF), classs proportionality (CPROP), class maximin share (CMMS), and utilitarian social welfare (USW). The authors' contributions offer nearly tight results for deterministic algorithms in both the context of indivisible and divisible item matching. While the authors achieve a 0.593-CPROP approximation guarantee for indivisible matching using a randomized algorithm, they do not address the challenge of simultaneously achieving a class envy-freeness (CEF) guarantee while ensuring non-wastefulness. This crucial open problem serves as the primary focus of our study. Furthermore, we explore various impossibility results for algorithms that align with the fairness-agnostic online matching literature, shedding light on the limitations and constraints of such approaches. While many other works examine online fair division [1, 8, 19, 39, 42], the majority restrict attention to additive valuations, rendering their techniques inapplicable to the matching setting. Finally, the recent work by [40] proposes a non-wasteful algorithm which guarantees CEF with high probability when the number of agents approach infinity.

**Price of Fairness** The first set of results on the Price of Fairness (PoF) traces back to [9] and [11]. [9] analyze the upper bound on the utility loss (specifically, egalitarian social welfare) incurred by fairness notions such as proportional fairness and max-min fairness in the allocation of divisible goods. A key takeaway from their results is that for a small number of players, the PoF remains relatively low; for example, for two players, the PoF for proportional fairness is at most 8.6%, and for max-min fairness, it is 11.1%. [11] further extend these results by examining fairness notions like proportionality, envy-freeness, and equitability for both divisible and indivisible goods and chores. However, as [7] highlight, a significant limitation in the indivisible setting is that the guarantees do not hold for every problem instance, as the results are not framed as a worst-case analysis. To address this, [7] investigate PoF under worst-case scenarios for various fairness criteria, including Nash social welfare, envy-freeness up to one good, balancedness, and egalitarian social welfare. It is important to note that these results do not directly apply to our setting, as the notion of class envy-freeness is not equivalent to any of the properties discussed above.

## 2 Model

For $t \in \mathbb{N}$, define $[t] = \{1, ..., t\}$. Consider a bipartite graph $G = (N, M, E)$ where $N$ represents a set of vertices henceforth referred to as "agents", $M$ a set of vertices called "items", and $E$ the set of incident edges. We say that $a \in N$ likes item $o \in M$ if the two are adjacent in $G$, i.e., $(a, o) \in E$. The set of agents $N$ is partitioned into $k$ (known) classes $N_1, ..., N_k$ such that $N_i \cap N_j = \emptyset$ for all $i \neq j$ and $\bigcup_{i=1}^{k} N_i = N$. We slightly abuse notation and call class $N_i$, class $i$.

**Matching.** We denote by the matrix $X = (x_{a,o})_{a \in N, o \in M} \in \{0, 1\}^{N \times M}$ a matching, where each $x_{a,o}$ indicates if item $o$ is matched to agent $a$. For divisible matchings, we replace $\{0, 1\}$ by $[0, 1]$. Given such a matching, we refer to an agent as *saturated* when $\sum_{o \in M} x_{a,o} = 1$ and an item as *assigned* if $\sum_{a \in N} x_{a,o} = 1$.

For some matching $X$, we let $Y(X) = \left(\sum_{a \in N_i} x_{a,o}\right)_{i \in [k], o \in M}$ be the matrix containing the items assigned to agents within each class. Further let $Y_i(X)$ denote the row of $Y(X)$ corresponding to class $i$. More specifically, this is the set of items matched to agents in class $i$:

$$Y_i(X) = \{o \in M : x_{a,o} = 1 \text{ for some } a \in N_i\}.$$

**Class Valuations.** For agent $a \in N$, the value of a matching $X$ is given by $V_a(X) = \sum_{o \in M:(a,o) \in E} x_{a,o}$ and we slightly abuse notation by defining the value of class $i$ from matching $X$ to be $V_i(X) = \sum_{a \in N_i} V_a(X)$. This is the so-called *utilitarian social welfare* (USW) of the matching for the agents in class $i$, and is equivalent to the standard matching size objective in the online bipartite matching literature.

Given a vector $\mathbf{y} = (y_o)_{o \in M} \in \{0, 1\}^M$ representing the allocation of different items, the *optimistic valuation*, $V_i^*(\mathbf{y})$, of class $i$ for $\mathbf{y}$ is the size of the maximum matching between the agents of $N_i$ and the items of $\mathbf{y}$. $V_i^*$ is equivalent to the maximum size of the integral matching between the items and agents in $N_i$ which can be computed with a standard LP – we defer the reader to [21] for more exposition on this definition. We emphasize that the optimistic valuation function is *subadditive*, and not additive as is a standard assumption leveraged in the fair division literature to obtain many algorithmic guarantees. As demonstrated in [21], it is exactly this functional property that prevents any deterministic algorithm for indivisible matchings from being non-wasteful and CEF1. Moreover, this aspect of the problem instance will necessarily make the analysis of our algorithmic guarantees and impossibility results non-trivial as compared to their additive counterparts.

## 2.1 Definitions of Fairness

The aim in the present work is to distribute the arriving goods among classes in a way that respects certain fairness criteria or principles. The commonly studied fairness criteria that we use here are envy-freeness [16], and proportionality [37].

For envy-freeness, we compare the value of $V_i(X)$ for the matched items to class $i$ and $V_i^*(Y_j(X))$, the optimistic value of class $j$'s matching according to $i$. Note that the optimistic valuation is necessarily larger than what could be obtained in the online model, so this is a particularly strong notion of fairness.

**Definition 2.1** (Class Envy-Freeness). A matching $X$ is $\alpha$-**class envy-free** ($\alpha$-CEF) if for all classes $i, j \in [k], V_i(X) \geq \alpha \cdot V_i^*(Y_j(X))$. For $\alpha = 1$, we simply call the matching **class envy-free** (CEF).

In general, CEF allocations cannot be guaranteed for indivisible matchings (ie. one item arrives to be distributed across two classes). We thus also consider the relaxed notion of class envy-freeness up to one item, consistent with the EF1 notion introduced by [30].

**Definition 2.2** (Class Envy-Freeness Up to One Item). A matching $X$ is $\alpha$-**class envy-free up to one item** ($\alpha$-CEF1) if for every pair of classes $i, j \in [k]$, either $Y_j(X) = \emptyset$ or there exists an item $o \in Y_j(X)$ such that $V_i(X) \geq \alpha \cdot V_i^*(Y_j(X) \setminus \{o\})$. When $\alpha = 1$, we simply refer to the matching as **class envy-free up to one item** (CEF1).

At the class level, the *proportional share* of class $i$ is defined as
$$\text{prop}_i = \max_{X \in \mathcal{I}} \min_{j \in [k]} V_i^*(Y_j(X))$$
where $\mathcal{I}$ is the set of (possibly divisible) matchings of the set of items $M$ to the set of agents $N$.

**Definition 2.3** (Class Proportional Fairness). We say that a matching $X$ is $\alpha$-**class proportional** ($\alpha$-CPROP) if for every class $i \in [k], V_i(X) \geq \alpha \cdot \text{prop}_i$. When $\alpha = 1$, we simply call this **class proportional** (CPROP).

We highlight that [21] demonstrate that any algorithm which assigns deterministically at the class level or within classes must be at best $\frac{1}{2}$-CPROP. Therefore, we must introduce randomness at both stages to surpass the performance of a deterministic algorithm. This further motivates the present studies' emphasis on randomized algorithms.

## 2.2 Definitions of Efficiency.

We consider two notions of efficiency. The first, non-wastefulness, ensures that no item is discarded if a matching is possible. In our integral assignment setting, non-wastefulness corresponds to a *maximal* matching.

**Definition 2.4** (Non-Wastefulness). We say that a matching $X$ is **non-wasteful** (NW) if there is no pair of agent $a$ and item $o$ such that $a$ likes $o$, $a$ is not saturated, and $o$ is not fully assigned.

The second efficiency measure is utilitarian social welfare which quantifies the size of the resultant matching. This is consistent with the classical objective for online matching when ignoring considerations of fairness.

**Definition 2.5** (Utilitarian Social Welfare). The **utilitarian social welfare** (USW) of a matching is given by $\text{usw}(X) = \sum_{a \in N} \sum_{o \in M : (a,o) \in E} x_{a,o}$. We say that a matching is $\alpha$-USW if $\text{usw}(X) \geq \alpha \cdot \text{usw}(X^*)$ for all matchings $X^*$. When $\alpha = 1$, we refer to $X$ as the USW-optimal matching.

It is well-known that maximal matchings (both divisible and indivisible) are a at least a $1/2$-approximation to the maximum. The proof of this fact is standard in the literature for maximum and maximal matchings, but is included in Appendix B for full clarity.

*Proposition* 2.6. Every non-wasteful matching is $\frac{1}{2}$-USW.

### 2.3 Online Model

In the online setting, the items in $M$ arrive one-by-one in an arbitrary order. We refer to the step in which item $o \in M$ arrives as step $o$. Upon the arrival of item $o$, the incident edges, $(a, o)$, to the agents in $a \in N$ are revealed from $G$. At this point, the algorithm must make an irrevocable decision to match the item one of the agents in $N$ who is not currently saturated (ie. not already included in the matching). We examine both deterministic (for analytic purposes) and randomized algorithms to construct these matchings.

In the online setting we define the online fairness metrics using the standard notion of a competitive ratio as our approximation factor as follows.

**Definition 2.7.** For $\alpha \in (0, 1]$, a deterministic online algorithm for matching items is $\alpha$-CEF (resp., CEF1, CPROP, USW, or NW) if it produces an $\alpha$-CEF (resp., CEF1, CPROP, USW, or NW) matching after all items have arrived.

For randomized algorithms, we require that all fairness constraints hold in expectation after all items have arrived.

## 3 Randomized Algorithms

We here present our randomized algorithm for constructing a matching that achieves simultaneous guarantees for the CEF and CPROP fairness objectives, as well as non-wasteful efficiency. While wastefulness makes the fairness objectives somewhat trivial to obtain, our enforced non-wasteful condition showcases the complexity of maintaining a fair matching. We begin by analyzing the simple random algorithm and later use this procedure to validate the impossibility result on the $\alpha$-CEF guarantee of any algorithm.

Our algorithm is a simple variant on a completely random matching procedure to ensure non-wasteful efficiency. Upon the arrival of an item $o$, it is revealed which classes $i \in [k]$ have a currently unmatched agent, $a$, that likes the item (ie. $(a, o) \in E$ and $x_a = 0$). Over this set of classes, we select one at random and then randomly assign the item to an unmatched agent that likes the item within this class. Though this nested randomization appears obvious, the proof of its near optimal fairness guarantees requires a nontrivial analysis. The following theorem establishes the approximate fairness and efficiency guarantees of our algorithm – the pseudocode is provided in Algorithm 1.

*Restatement of Theorem* 1.1 (Formal). For randomized matching of indivisible items, Algorithm 1 satisfies non-wastefulness, $\frac{1}{2}$-CEF, $\frac{1}{2}$-CPROP, and $\frac{1}{2}$-USW

*Proof Sketch.* We here provide a sketch of the analysis needed to verify the result and defer the reader to Appendix B.2 for the complete analysis.

---

**Algorithm 1** RANDOM

---
1: **for** $o \in M$ **do**
2:     $S_o \leftarrow \emptyset$
3:     **for** $i \in [k]$ **do**
4:         **if** $\exists a \in N_i$ s.t. $(a, o) \in E$ and $x_a = 0$ **then**
5:             $S_o \leftarrow S_o \cup \{i\}$
6:         **end if**
7:     **end for**
8:     Pick an $i \in S_o$ uniformly at random
9:     Pick an $a \in N_i$ with $(a, o) \in E$ and $x_a = 0$ uniformly at random
10:     Set $x_{a,o} = 1$
11: **end for**

---

The non-wastefulness of the algorithm is direct from its definition: each arriving item is allocated to an unsaturated agent that likes the item. The USW approximate guarantee then follows from Proposition 2.6.

For the CEF objective, we invoke a novel proof technique specific to the challenging analysis of optimistic valuations used throughout the fair matching objective framework. Specifically, we show that for any two distinct classes $i, j$, the expected value $\mathbb{E}[V_i(X)] \geq \frac{1}{2} \cdot \mathbb{E}[V_i^*(Y_j(X))]$ by introducing dummy items to analyze the value of some augmented input set $A_i$, proving that $V_i^*(A_i) \leq 2 \cdot V_i(X)$. With this augmented set, we more readily obtain the desired approximate guarantees and demonstrate that the optimal solution on $A_i$ dominates the true solution on $Y_j(X)$. Combining this fact with a lemma relating the expected values, we establish the $\frac{1}{2}$-CEF guarantee. We suspect this novel proof technique will be useful in follow up works that examine similar, nonadditive, solution concepts – a key problem identified by the preliminary work of [21].

Lastly, for the CPROP objective, the analysis modifies the Equal-Filling-OCS approach of [21] by using a simpler independent rounding method. More concretely, for an arriving item $o \in M$ we construct the vector $(x_{a,o})_{a \in N}$ where each entry is the corresponding probability of an agent being matched to the given item under the RANDOM algorithm. After all items have arrived, each agent $a$ is matched to an item with probability

$$1 - \prod_{o \in M}(1 - x_{a,o}) \geq 1 - \exp\left(-\sum_{o \in M} x_{a,o}\right)$$

and by integrating according to this density function over the agents in each class and comparing to the upper bound on the $\text{prop}_i$ value from [21], we obtain the desired approximation ratio. $\square$

We here highlight that while the more sophisticated OCS rounding scheme and, as a result, the Equal-Filling-OCS algorithm yields a stronger 0.593-CPROP guarantee, the algorithm does not lend itself to analysis of the CEF objective (it remains an open problem to obtain any such approximation for the algorithm). Thus, although our algorithm is slightly weaker with respect the CPROP fairness definition, our simple algorithm further gives a $\frac{1}{2}$-CEF approximation while maintaining non-wastefulness.

## 4   Improved CEF Upper Bounds

In this section, we provide improved CEF upper bounds of 0.761 for any randomized indivisible and 0.677 for any deterministic divisible algorithm.

### 4.1   Indivisible setting

Our upper bound for the indivisible setting showcases the near tightness of our algorithmic guarantees proven in Section 3.[2]

The seminal paper of [28] showed that no online algorithm can get a competitive ratio better than $1 - 1/e$ for the USW objective by an "upper triangular graph" (the graph whose adjacency matrix is upper triangular) construction. In this graph, there are $n$ arriving items and $n$ agents. The first item to arrive has an edge to all $n$ agents, the second has an edge to $n - 1$ of the agents, the third to only $n - 2$ of them, and so on.

We will proceed with demonstrating that by an extension on the result of [28], we can upper bound the $\alpha$-CEF guarantee of any randomized algorithm. Specifically, we prove the following theorem:

*Restatement of Theorem* 1.2 (Formal). No randomized online algorithm for matching indivisible items can achieve an $\alpha$-CEF guarantee for any $\alpha > \left(\frac{e^2-1}{e^2+1}\right)$ and non-wastefulness.

Our construction for CEF impossibility extends the problem instance of [28] to include a second class which can be uniquely matched to each arriving item (See Figure 2a for an illustration of

---

[2]We note that, trivially, an upper bound of $1 - 1/e$ exists for the USW objective by the classic result of [28]. This bound persists for CPROP by considering the problem instance where $k = 1$.

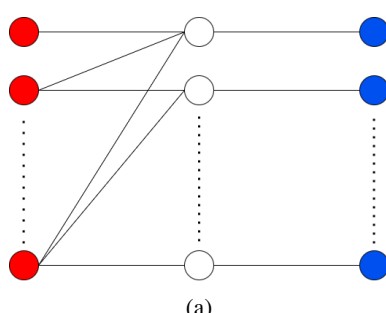 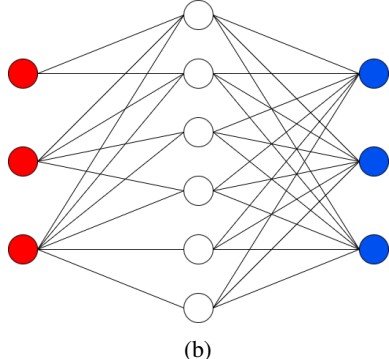

| (a) | (b) |

Figure 2: Impossibility constructions for the upper bound results of Theorems 1.2 and 1.3. (a) the indivisible setting construction for an at most $\left(\frac{e^2-1}{e^2+1}\right)$-CEF approximation, (b) the divisible setting construction for an at most 0.677-CEF approximation.

this instance). We proceed to show that this instance admits at best a $\frac{e^2-1}{e^2+1}$ approximation to the CEF objective by first showing that the RANDOM algorithm admits this approximation factor in expectation and subsequently showing that no algorithm can do better on the given instance, thus bounding the performance of any randomized algorithm in general.

Most crucial to the argument that bounds $\alpha$ is the computation of a "stopping time" after which no further items can be matched to Class 1 since all potential agents will have been previously saturated in a suboptimal manner – a result of the ambiguity in matching from the upper triangular instance and randomization of our algorithm. After this stopping time, by non-wastefulness, all items will be matched to the second class. This necessarily results in an unfair distribution of items and by deriving the stopping time as a fraction of the number of items, we obtain our upper bound approximation.

The full proof of this theorem is deferred to Appendix B.3 due to space constraints.

## 4.2 Divisible setting

In this section, we improve upon the established bounds of [21] and move closer towards resolving the open problem on what is the best achievable $\alpha$-CEF guarantee for non-wasteful divisible matchings through a novel input construction. Prior to this work, the best known upper bound bound was $3/4$ with an algorithm that guarantees at least $1 - 1/e$. Our improved bound of $0.677$ is thus nearly tight. Note that our CEF upper bound is subject to non-wastefulness because an algorithm can trivially achieve fairness on its own by throwing away every item.

**Theorem 4.1.** *No non-wasteful deterministic algorithm for matching divisible items can achieve $\beta$-CEF for any $\beta > 0.677$.*

*Proof Sketch.* Due to space constraints, the full proof is found in Appendix B.3. We here briefly give the intuition for the impossibility result.

We construct an adversarial instance for which we cannot achieve $\beta$-CEF for $\beta > 0.677$. Consider two classes, each comprised of $n$ agents, and an input stream of $2n$ items. For the first class, items numbered 1 and 2 are connected to all the agents. And for each $i$, items numbered $2i + 1$ and $2i + 2$ are connected to all the agents that items $2i - 1$ and $2i$ are, except one arbitrary agent from the class. Therefore, items $2i - 1$ and $2i$ are connected to $n - i + 1$ agents for each $i$. For the second class, we simply have a complete graph wherein each agent of the class has an edge connecting her to each item. The full construction for $n = 3$ case is depicted in Figure 2b for clarity.

To verify the result for the given instance, assume there is an algorithm that guarantees $\alpha$-CEF. The proof then follows from the synthesis of two key facts: (i) any $\alpha$-CEF algorithm should distribute items equally among agents within Class 1 for this input instance to maximize their saturation (Lemma B.7), and (ii) that any such algorithm must further divide arriving items between the two classes such that Class 1 receives $\frac{1+\alpha}{2}$ and Class 2 receives $\frac{1-\alpha}{2}$. This ratio is optimal against an adversarial input, ensuring neither class is overly envious (Lemma B.8).

The first result is proven by the nature of a non-wasteful online algorithm, and the second is achieved by induction: assume the $\alpha$-CEF guarantee holds up to step $t-1$. For step $t$, if the item is not allocated according to the prescribed ratio, an adversary can force a violation of the $\alpha$-CEF guarantee. Thus, maintaining the ratio ensures the guarantee persists.

The theorem is finally proven by bounding the matching size for each class given the two above facts. We can ultimately determine the maximum value of $\alpha$ that maintains the $\alpha$-CEF guarantee, concluding that $\alpha \leq 0.677$. □

## 5 Price of Fairness

Stemming from the highly influential work of [28], it is well-known that no algorithm for the online matching problem can achieve an approximation better than $1 - 1/e$ to the maximal matching objective (USW in our context). Moreover, by the result of Theorem 1.2, we demonstrate that a comparable approximation bound persists for the CEF objective. It is, thus, only natural to explore the following question: *is it possible to achieve both an optimal CEF and USW approximation simultaneously?*

We here address this question with an impossibility result that provides an initial trade-off between the fairness of a solution and its optimality with respect to the (unfair) USW objective – a relationship we refer to as the *price of fairness* for online matching. More formally, our result is as follows.

**Theorem 5.1.** *For any $\varepsilon > 0$, there exists a problem instance such that no (possibly randomized) non-wasteful online algorithm with an $\alpha$-CEF guarantee can achieve an approximation to the USW objective greater than $\frac{1}{1+\alpha} + \varepsilon$.*

*Proof Sketch.* The proof proceeds by considering an instance with $k-1$ classes of $q$ agents, as well as a $k$-th class comprised of $q(k-1)$ agents. The adversarial input stream consists of two phases. In the first phase, $p(k-1) + q$ items arrive, each of which has incident edges to every agent in the graph, whereas in the second phase, $k-1$ groups of items arriving sequentially where the $i$-th such group consists of $q$ items with edges to all the agents in class $i$.

In this instance, we first show that any non-wasteful $\alpha$-CEF algorithm should allocate at least $p(k-1)$ items to the class $1, 2, \ldots, k-1$. Then, since these items cannot be further matched to class $1, \ldots, k-1$ in the second phase, we can upper bound the utilitarian social welfare at the end of the second phase by $p(k-1) + qk - p(k-1) = qk$. Observing that the offline optimal solution in hindsight allocates all items in the first phase to class $k$, while allocating the remaining $q$ items specific to each class to their corresponding class in the second phase, the optimal USW is $p(k-1) + q + q(k-1)$. The proof follows from selecting proper parameters for $p$ and $q$. □

With this novel price of fairness result, we observe that if $\alpha > \frac{1}{e-1} \approx 0.582$, then our result for $\alpha$-CEF implies a USW guarantee that is strictly smaller than $1 - \frac{1}{e}$, the well known bound by [28]. Thus, USW must be sacrificed to achieve fairness. Further, if there exists any algorithm that achieves the 0.761-CEF guaranteed by Theorem 1.2, it would necessarily have USW smaller than 0.568 (larger than 0.5 by maximality), which is considerably smaller than the $1 - \frac{1}{e} \approx 0.632$ by [28].

## 6 Discussion & Open Problems

Our work closes the long-standing open conjecture on whether a non-wasteful randomized algorithm can achieve non-trivial fairness guarantees in the context of online matching problems. By conducting a detailed and non-trivial analysis of a natural randomized matching procedure, we have successfully developed an algorithm that not only complements our previously established 0.76-CEF upper bound construction but also aligns with the known upper bounds for the CPROP and USW efficiency guarantees.

Moreover, we demonstrate that the algorithmic guarantee of [21] for deterministic and divisible matchings is almost tight, with a novel input construction that exhibits a 0.67-CEF upper bound. We lastly define "price of fairness" for the online matching problem and present an interesting impossibility result on the trade-off between fair and optimal solutions. Namely, we demonstrate that any approximation to the CEF objective follows an inverse proportionality relationship to the

possible USW approximation any algorithm can obtain. This demonstrates that we must allow some degradation in solution quality to ensure equitable treatment of classes.

Future work should address the natural gaps between the upper and lower bounds discussed above. Perhaps most importantly, can a randomized algorithm achieve a USW approximation better than $\frac{1}{2}$ while maintaining the given fairness guarantees? Is the price of fairness result tight, ie. does an algorithm exist that ensures $\alpha$-CEF while simultaneously guaranteeing $\frac{1}{1+\alpha}$-USW? We believe that some of these answers may result from a careful extension on the RANKING algorithm that will naturally rely on some priority ordering over both classes and agents.

# 7 Acknowledgements

This work is partially supported by DARPA QuICC, NSF AF:Small #2218678, NSF AF:Small #2114269, Army-Research Laboratory (ARL) #W911NF2410052, and MURI on Algorithms, Learning and Game Theory. Max Springer was supported by the National Science Foundation Graduate Research Fellowship Program under Grant No. DGE 1840340. Any opinions, findings, and conclusions or recommendations expressed in this material are those of the author(s) and do not necessarily reflect the views of the National Science Foundation.

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

## A  Fairness and Efficiency in Online Class Matching

We here extend the classical notion of Nash social welfare (NSW) to the class fair matching problem setting and demonstrate its intersection with the notion of CEF1. Concretely, we demonstrate that the former implies the latter, but a reverse implication is not guaranteed.

**Definition A.1** (Class Nash Social Welfare). A class Nash social welfare (*CNSW*) of a matching $X$ is defined by

$$\text{CNSW(X)} = \left( \prod_{i=1}^{k} V_i(X) \right)^{1/k}.$$

We say that a matching is *CMNW* if it maximizes *CNSW*.

The NSW has been typically viewed as a balance between fair and efficient allocations of items to a set of agets. Most recently, [41] showed the NSW objective is the only welfarist function of agent valuations whose maximization leads to allocations that are EF1 and PO. In spite of its tremendous value, it is also widely known that maximizing the objective is generally hard even to approximate within polynomial time without strong problem assumptions [6]. We here demonstrate that, given such a maximizing matching for the CNSW objective, we further obtain the CEF1 guarantee in line with the fair division literature. For a more complete discussion on the NSW objectives role in fair division, we defer the reader to [2].

**Theorem A.2.** *A* CMNW *matching satisfies non-wastefulness and is* CEF1.

*Proof.* Consider a matching $X$ that maximizes the CNSW objective. We first note that any wasteful matching can necessarily increase the CNSW objective by matching any wasted item, so we have non-wastefulness must hold.

Now, towards contradiction, suppose that $X$ does not satisfy CEF1. By the definition of CEF1, this implies that there exists two classes $i, j \in [k]$ such that

$$V_i(X) < V_i^*(Y_j(X) \setminus \{o\}),$$

for some $o \in Y_j(X)$. Let $V_i(X) = |X_i| = s$. By the assumption above, we must have

$$V_i^*(Y_j(X) \setminus \{o\}) > V_i(X) = |X_i| = s,$$

and thus $|Y_j(X) \setminus \{o\}| \geq s + 1$ for some $o \in Y_j(X)$ since $s$ is an integer. Combining this result with the non-wastefulness of a CNSW maximizing matching, we further have that $V_j(X) = |X_j| = |Y_j(X)| \geq s + 2 > V_i(X) + 1$. We now verify the following claim that will be crucial in proving the final CEF1 result.

*Claim* A.3. There exists an item $o' \in Y_j(X)$ such that

$$V_i(X_i \cup \{o'\}) > V_i(X_i)$$

Before proving this claim, we show how it yields the CEF1 guarantee: starting from matching $X$, consider a modified allocation $X'$ that moves the item $o' \in Y_j(X)$ from Claim A.3 to $X_i$. We compute the CNSW of this modified matching as

$$\text{CNSW}(X') = \left( (V_i(X) + 1) \cdot (V_j(X) - 1) \prod_{[k] \ni s \neq i, j} V_s(X) \right)^{1/k}$$

$$\geq \left( V_i(X) \cdot V_j(X) \prod_{[k] \ni s \neq i, j} V_s(X) \right)^{1/k}$$

$$= \text{CNSW}(X)$$

where the inequality follows from the contradictory assumption. Naturally, this contradicts the maximality of $\text{CNSW}(X)$, thus we must have that $X$ is a CEF1 matching. ☐

We conclude the theorem's proof by verifying Claim A.3.

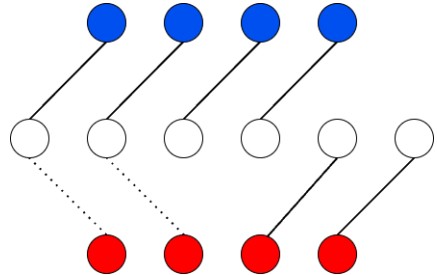

Figure 3: Hardness instance for Theorem A.4.

*Proof of Claim A.3.* Fix an item $o \in Y_j(X)$ and let $S_i$ denote the set of agents in class $i$ who are matched under $X$. We similarly define $\hat{S}_i^*$ be the set of saturated agents in class $i$ under the optimistic matching of items in $Y_j(X) \setminus \{o\}$. Since $V_i^*(Y_j(X) \setminus \{o\}) > Y_i(X)$, we have $|S_i^* \setminus S_i| \geq 1$.

Now, pick any agent $a' \in S_i^* \setminus S_i$, and let $o'$ denote the item that is matched to $a'$ under the optimistic matching on $Y_j(X) \setminus \{o\}$. By the selection of $a'$ and $o'$, we must have that $o'$ can be matched to agent $a'$ without any conflicts. Therefore, $V_i(Y_i(X) \cup \{o'\}) > V_i(X)$. $\qquad \square$

We demonstrate that although the CMNW matching provides the favorable CEF1 fairness guarantee, the opposite relationship is not necessarily ensured. The following theorem formalizes this notion that CMNW is a strictly stronger property than CEF, which itself is stronger than CEF1.

**Theorem A.4.** *A non-wasteful CEF matching is not guaranteed to maximize CNSW.*

*Proof.* Consider two classes $N_1$ and $N_2$, each of which consists of four agents, denote these agents by $a_1, a_2, a_3, a_4$ and $b_1, b_2, b_3, b_4$ respectively. Fix an adversarial sequence of six items indexed by $1, 2, \ldots, 6$. Items $1, 2, 3$ and $4$ have edges with all the agents in class $N_2$, and item $1$ further has an edge with agent $a_1$. Items $5$ and $6$ have edges with agents $a_3$ and $a_4$. Consider a matching $X$ that matches items $1, 2, 3, 4$ to class $N_2$ and $5, 6$ to class $N_1$ (See Figure 3 for a depiction). Due to the construction of edges, this must be a non-wasteful matching and the NSW is computed to be $\sqrt{2 \cdot 4}$. Note further that allocating items $1, 2, 3, 4$ to class $N_1$ only induces a matching to class $N_2$ of optimistic value $V_2(Y_1^*(X) = 2$, and thus $X$ is CEF. On the flip side, if we consider an allocation $X'$ that allocates items $2, 3, 4$ to class $N_2$ and $1, 5, 6$ to class $N_1$, this also constitutes a non-wasteful matching and the NSW is $\sqrt{3 \cdot 3} > \sqrt{2 \cdot 4}$. Therefore, a non-wasteful CEF matching does not necessarily maximize the CNSW objective. $\qquad \square$

# B  Omitted Proofs

## B.1  Proofs from Section 2

*Proof of Proposition 2.6.* Let $X^*$ be a matching that maximizes the USW objective, and let $X$ be any non-wasteful matching in the same instance. By definition of non-wastefulness above, we must have that each edge in the graph $G$ has at least one end point included in the matching, ie. for every $(a, o) \in E$, $\sum_{o' \in M} x_{a,o'} = 1$ or $\sum_{a' \in N} x_{a',o} = 1$. Therefore, we have

$$\text{usw}(X^*) = \sum_{a \in N} \sum_{o \in M} x_{a,o}^*$$

$$\leq \sum_{(a,o):x_{a,o}^*=1} \left( \sum_{a' \in N} x_{a',o} + \sum_{o' \in M} x_{a,o'} \right)$$

$$\leq \sum_{a \in N} \sum_{o' \in M} x_{a,o'} + \sum_{a' \in N} \sum_{o \in M} x_{a',o}$$

$$= 2 \cdot \text{usw}(X)$$

where the first inequality comes from the fact that $x_{(a,o)}^* = 1$ implies at least one of the end points for each $(a, o)$ included in the maximum maching must be in the non-wasteful matching. By the

summation properties on edges in a non-wasteful matching we obtain the final equality and, therefore, a $\frac{1}{2}$-USW approximation. $\square$

## B.2 Proofs from Section 3

*Proof of Theorem 1.1.* The non-wastefulness of the algorithm follows immediately from its definition: at each step, the arriving item is allocated to an unsaturated agent that likes the item. Also, USW immediately follows from Proposition 2.6. Thus, in what follows, we focus on proving CEF and CPROP guarantees of the algorithm.

**CEF.** Let $X$ be the matching constructed by the randomized algorithm after the arrival of all items in $M$. Consider any two distinct classes $i, j \in [k]$. We seek to prove that

$$\mathbb{E}\left[V_i(X)\right] \geq \frac{1}{2} \cdot \mathbb{E}\left[V_i^*(Y_j(X))\right].$$

According to our algorithm, for any item $o$ liked by at least one agent in the class $i$, the probability of allocating this item to $i$ is at least that of allocating to another class $j$ unless, at step $o$, all agents who like the arriving item are already saturated.

For ease of analysis, we consider a simple upper bounding scheme that leverages the above fact. Specifically, upon the arrival of item $o$ that is liked by at least one agent in $i$ but all such agents are saturated, we create a dummy item $\bar{o}$ that is identical to $o$ and add it to the set of matched items to class $i$. We therefore augment the set $X_i$ to include these dummy items which we denote as the set $A_i \supseteq X_i$. Using this, we prove the following claim.

*Claim* B.1. The optimistic value of set $A_i$ to class $i$, denoted as $V_i^*(A_i)$, is at most twice the size of its matching in $X$. More formally:

$$V_i^*(A_i) \leq 2 \cdot V_i(X).$$

*Proof.* We first observe that if $A_i = X_i$, then $V_i^*(A_i) = V_i^*(X)$ and since the optimistic value constructs a maximal matching, the claim holds by application of Proposition 2.6.

We therefore assume that $X_i \subsetneq A_i$. By the order of arrival of items, we must have that $V_i(X) = V_i(A_i)$ since the items in $A_i \setminus X_i$ can only be matched to agents that are currently saturated. However, under the optimistic valuation, the matching can only increase in size, i.e., $V_i^*(X) \leq V_i^*(A_i)$. Again, by application of Proposition 2.6 and the maximality of the optimistic value of $A_i$, we must have that

$$V_i^*(A_i) \leq 2 \cdot V_i(A_i) = 2 \cdot V_i(X)$$

completing the proof of the claim. $\square$

We lastly need the following lemma to proof the main theorem.

*Lemma* B.2. For any two distinct classes $i, j \in [k]$, $\mathbb{E}\left[V_i^*(A_i)\right] \geq \mathbb{E}\left[V_i^*(Y_j(X))\right]$ where expectations are taken over the randomness of Algorithm 1.

The combination of this lemma with Claim B.1, we have that

$$\mathbb{E}\left[V_i(X)\right] \geq \frac{1}{2} \cdot \mathbb{E}\left[V_i^*(A_i)\right] \geq \frac{1}{2} \cdot \mathbb{E}\left[V_i^*(Y_j(X))\right]$$

thus, we have the $\frac{1}{2}$-CEF guarantee and have proven the theorem.

It therefore remains to prove Lemma B.2.

*Proof of Lemma B.2.* To prove the lemma, we need to verify two essential claims.

*Claim* B.3. For an arbitrary item, $o$, liked by an agent in class $i \in [k]$, the probability that $o$ (or its copy) is in $A_i$ is greater than or equal to the probability that $o \in Y_j(X)$ for $i \neq j \in [k]$.

*Proof.* If there exists an agent in class $i$ that likes $o$ and is unsaturated at the time of its arrival, then the item will be added to $X_i$ with equal probability to any other class $j$ containing such an agent, so the claim holds.

If instead all such agents in class $i$ are saturated, then a copy of $o$ will be added to $A_i$ with probability 1. Thus, we have the claim. $\square$

We now leverage this claim to prove a slightly stronger result which will ultimately yield the lemma result.

*Claim* B.4. For an arbitrary item $o$ liked by an agent of class $i$, we must have that at least one of the following properties hold:

1. for $j \neq i \in [k]$, $\mathbf{Pr}\left[o \in Y_j(X)\right] = 0$

2. $\mathbf{Pr}\left[o \in A_i\right] = 1$

3. or $\mathbf{Pr}\left[o \in Y_j(X)\right] = \mathbf{Pr}\left[o \in A_i\right]$

*Proof.* Upon the arrival of $o \in M$, there are three possibilities for its irrevocable assignment. If there is no agent in $j$ that likes $o$, we must have that $\mathbf{Pr}\left[o \in Y_j(X)\right] = 0$. Alternatively, if such an agent exists and is currently unsaturated but all the viable agents in $i$ are saturated, we must have that $o$ is added to $A_i$. Lastly, if both classes have unsaturated agents that like the item, then either $Y_j(X)$ or $X_i \subseteq A_i$ will receive the item with equal probability. Thus, we have the claim. $\square$

We now have the necessary tools to argue about the expected optimistic values of the sets $A_i$ and $Y_j(X)$ according to class $i$. Clearly, only items in $Y_j(X)$ that are liked by agents of class $i$ contribute to the optimistic valuation. By the above claims, each such item is allocated to $A_i$ with greater than or equal to the probability that they are allocated to $Y_j(X)$. We therefore have the result by linearity of expectations for these item assignments. $\square$

**CPROP.** The analysis of our CPROP guarantee relies on a modification to that of Equal-Filling-OCS from [21].

*Proof.* Let $f(x)$ denote the number of agents who are matched at least $x \in [0, 1]$ (where $x = 1$ would imply the agent is saturated) under the divisible matching that corresponds to the probabilities from RANDOM. At each step of the online input stream, an item $o \in M$ arrives and the random algorithm produces a vector $(\tilde{x}_{a,o})_{a \in N}$ of probabilities for selecting each agent as follows: if an agent does not like the item, then $x_{a,o} = 1$, otherwise we set this value to be the probability of its selection by RANDOM. We then select an agent to be matched to the item with probability $\tilde{x}_{a,o}$.

By the end of the item arrivals, each agent is selected to match to an arriving item with probability

$$1 - \prod_{o \in M} (1 - \tilde{x}_{a,o}) \geq 1 - e^{-\sum_{o \in M} \tilde{x}_{a,o}}.$$

We henceforth denote the above probability bound by $p(\tilde{x}_a)$ for brevity. We can therefore bound the expected value of the matching to a class $i$ by:

$$\mathbb{E}\left[V_i(X)\right] \geq \sum_{a \in N_i} p(\tilde{x}_a)$$

$$= -\int_0^\infty p(\theta) df(\theta)$$

$$= \int_0^\infty p'(\theta) f(\theta) d\theta$$

where the first equality comes from the definition of $f(\theta)$ and the last from an integration by parts.

As noted in [21], we have that for any divisible matching $\tilde{X}$ denoted by $\tilde{Y}_i$ for $i \in [k]$ such that $\sum_{i \in [k]} \tilde{Y}_{i,o} = 1$ for each $o \in M$:

$$\sum_{j \in [k]} V_i^*(\tilde{Y}_j) \leq k \cdot \left( \int_0^\theta f(z)dz + f(\theta) \right) \quad \forall \theta > 0.$$

Therefore, the $\mathrm{prop}_i$ value (which is a maximization over *divisible matchings*) can be upper bounded as

$$\mathrm{prop}_i \leq \int_0^\theta f(z)dz + f(\theta)$$

for all $\theta > 0$. We can further multiply this bound by non-negative coefficients and integrate with respect to $\theta$ to obtain

$$\mathrm{prop}_i \cdot \int_0^\infty c(\theta)d\theta \leq \int_0^\infty c(\theta) \left( \int_0^\theta f(z)dz + f(\theta) \right) d\theta$$

$$= \int_0^\infty c(\theta) \int_0^\theta f(z)dzd\theta + \int_0^\infty c(\theta)f(\theta)d\theta$$

$$= \int_0^\infty \left( \int_z^\infty c(\theta)d\theta + c(z) \right) f(z)dz$$

If we now choose the coefficients so that the relation $\int_z^\infty c(\theta)d\theta + c(z) = p'(z)$ holds for all $z > 0$, we obtain that

$$\mathrm{prop}_i \cdot \int_0^\infty c(\theta)d\theta \leq \int_0^\infty p'(z)f(z)dz \leq \mathbb{E}\left[ V_i(X) \right].$$

We lastly obtain the approximation factor by directly computing the integral

$$\int_0^\infty c(\theta)d\theta = \int_0^\infty -e^\theta \int_\theta^\infty p''(y)e^{-y}dyd\theta$$

$$= \int_0^\infty e^\theta \int_\theta^\infty e^{-2y}dyd\theta$$

$$= \frac{1}{2} \int_0^\infty e^{-\theta}d\theta = \frac{1}{2}.$$

Thus, we have the result. □

□

## B.3 Proofs from Section 4

### B.3.1 Indivisible Setting

*Proof of Theorem 1.2.* Let $\tau$ denote the step in the input stream where no further items can be matched to the first class and note that $\tau \geq \frac{n}{2}$. Further note that, after step $\tau$, all items will be matched to class 2. Let $n_1(t)$ be the random variable denoting the number of available agents in class 1 for the item arriving at time $t$ and let $x(t)$ denote the number of items remaining in the input stream. We additionally let $\mathrm{OPT}_t$ denote the optimal matching agent in class 1 at time $t$. Further denote $\Delta n_1 = n_1(t) - n_1(t-1)$ and $\Delta x = x(t) - x(t-1)$. Naturally, $\Delta x = -1$ after every iteration, but for the $n_1(t)$ value we must consider three potential scenarios:

- $\Delta n_1 = 0$ : this occurs when $\mathrm{OPT}_t$ is already saturated and the arriving item is matched to class 2.

- $\Delta n_1 = -2$ : this occurs when $\mathrm{OPT}_t$ is unsaturated and the arriving item is not optimally matched to this agent.

- $\Delta n_1 = -1$ : this occurs in all other events.

Using these facts, we obtain the following lemma.

*Lemma* B.5. In the setting of the proof of Theorem 1.2, the expected value of $\tau$ is $\frac{n}{e^2} + o(n)$.

Before proving the lemma, we demonstrate how it is used in conjunction with the optimality of RANDOM for the given instance to complete the theorem's proof. Observe that

$$\mathbb{E}\left[V_1(X)\right] = \sum_t \mathbf{Pr}\left[o_t \text{ matched to class 1}\right]$$

$$= \sum_t \frac{1}{2}\mathbf{Pr}\left[n_1(t) \neq 0\right]$$

$$= \frac{1}{2}\sum_t \mathbf{Pr}\left[n_1(t) > 0\right]$$

$$= \frac{1}{2}\mathbb{E}\left[\tau\right]$$

where the second equality draws from the fact that, while there is an available matching in class 1, an item has a $\frac{1}{2}$ probability of being matched to that class. Lastly, combining with the result of Lemma B.5 we have

$$\mathbb{E}\left[V_1(X)\right] = \frac{1}{2}\left(n - \frac{n}{e^2} - o(n)\right)$$

with the remaining items going to class 1. Comparing the two class matching sizes obtains the desired bound of $\frac{1-1/e^2}{1+1/e^2} \approx 0.762$.

Lastly, by invoking the final key lemma below, we obtain the result.

*Lemma* B.6. The CEF performance of any non-wasteful online matching algorithm is upper bounded by the expected size of the matching produced by the RANDOM algorithm on the instance of instance $(I, \pi)$.

Thus, the competitive ratio proved above is the best achievable for the given instance and we are done. □

It remains to verify the two crucial lemmas leveraged in the proof above.

*Proof of Lemma B.5.* We proceed computing the expected value of $\Delta n_1$ under two different conditions: $\mathrm{OPT}_t$ being saturated or not.[3]

First, assume $\mathrm{OPT}_t$ is saturated at some earlier iteration. Then, with probability $\frac{1}{2}$ the item arriving at time $t$ is matched to class 2 and $\Delta n_1 = 0$, otherwise we must have that $\Delta n_1 = -1$. Therefore, we obtain

$$\mathbb{E}\left[\Delta n_1 | \mathrm{OPT}_t \text{ saturated}\right] = \frac{1}{2}(-1) + \frac{1}{2}(0) = -\frac{1}{2}.$$

Next, we assume $\mathrm{OPT}_t$ is unsaturated. Again, with probability $\frac{1}{2}$ the arriving item is matched to class 2 and we decrease by -1. If, instead, the item is matched to class 1 then $\Delta n_1$ can be either -1 or -2 depending on how the item is matched. Since at time $t$ there are $n_1(t)$ potential agents in class 1, the probability that the item is matched optimally is $\frac{1}{n_1(t)}$. Thus, we have

$$\mathbb{E}\left[\Delta n_1 | \mathrm{OPT}_t \text{ unsaturated}\right] = \frac{1}{2}(-1) + \frac{1}{2}\left(\frac{1}{n_1(t)}(-1) + \left(1 - \frac{1}{n_1(t)}\right)(-2)\right) = \frac{1}{2}\left(\frac{1}{n_1(t)} - 3\right).$$

It remains to compute the probability that $\mathrm{OPT}_t$ is saturated. Note that by construction of our instance and the random algorithm, the probability that $\mathrm{OPT}_t$ is unsaturated by time $t$ is exactly equal to the number of $n_1(t)$ sized subsets of $\{1, \ldots, x(t)\}$ which include the optimal agent [28]. Thus,

$$\mathbf{Pr}\left[\mathrm{OPT}_t \text{ unsaturated}\right] = \frac{n_1(t)}{x(t)}.$$

---

[3]Note that we are also implicitly assuming throughout this argument that $n_1(t) > 0$, i.e., at least one agent is still unsaturated in class 1.

We can now finally compute that

$$\mathbb{E}\left[\Delta n_1\right] = -\frac{1}{2} \cdot \left(1 - \frac{n_1(t)}{x(t)}\right) + \frac{1}{2} \cdot \frac{n_1(t)}{x(t)} \left(\frac{1}{n_1(t)} - 3\right).$$

Rearranging and simplifying we obtain that

$$\mathbb{E}\left[\Delta n_1\right] = -\frac{1}{2}\left(1 + \frac{2n_1(t) - 1}{x(t)}\right) \Rightarrow \frac{\mathbb{E}\left[\Delta n_1(t)\right]}{\Delta x(t)} = \frac{1}{2}\left(1 + \frac{2n_1(t) - 1}{x(t)}\right)$$

and by Kurtz' theorem [29], this is closely approximated by the following differential equation with probability tending to 1 as $n$ tends to infinity:

$$\frac{dn_1}{dx} = \frac{1}{2}\left(1 + \frac{2n_1 - 1}{x}\right)$$

Solving with the initial condition that $n_1 = x = n$ and setting the resultant equation equal to 0, we obtain that the expected stopping time is $\tau = n(1 - \frac{1}{e^2}) - o(n)$. □

*Proof of Lemma B.6.* We first note that any item in instance $(I, \pi)$ allocated to class 2 can only be matched to one specific agent, so there is no decision to be made for items in this class. Moreover, for items that are allocated to class 1, the best possible matching is achieved in expectation by allocating completely at random to the potential agents, as proven in [28]. We therefore need only prove that RANDOM is optimal at the *class* level. We proceed to verify this by comparing with a general algorithm representative of the other possible class matchings.

For any arriving item, if only one class has a potential matching then by non-wastefulness we must give the item to that class. Therefore, assume item $o \in M$ arrives and can be matched to either class (i.e., both have a potential matching to a currently unsaturated agent). However, from the perspective of the algorithm, there is no way of distinguishing the two classes and any bias towards one can be exploited by the adversary by flipping the given problem instance. Thus, the best we can do is randomly pick one of the two classes to match the item and we have the result of the lemma. □

### B.3.2 Divisible Setting

Assume we have an algorithm that guarantees $\beta$-CEF.

*Lemma* B.7. To ensure a maximal number of agents in Class 1 are saturated, it is optimal to distribute arriving items equally among the agents in this class.

*Proof of Lemma B.7.* For some $i$, consider the associated items $2i - 1$ and $2i$. We argue that it is optimal to distribute these two items equally within Class 1.

By the adversarial construction of the adjacency in Class 1, we know that one of the agents who likes these items does not like any of the following items. In the offline setting, it is straightforward to match the items to their associated agent and ensure a perfect matching. However, in the online setting, we do not know which of the current agents will be unavailable for future matching in the adversarial input stream. Therefore, to in maximizing the USW, it is optimal to distribute this item equally within the first class. □

Now, let $\alpha = \frac{1-\beta}{1+\beta}$ and reciprocally define $\beta = \frac{1-\alpha}{1+\alpha}$. We proceed to demonstrate that any $\beta$-CEF algorithm must allocate arriving items in a strategic manner between the two classes of the given adversarial input to maintain this approximate guarantee.

*Lemma* B.8. Upon arrival of item $o_t$, if there exists $a \in N_1$ such that $\sum_{k<t} x_{a,o_k} < 1$ then any $\beta$-CEF algorithm must divide $o_t$ among the two classes such that the first class receives $\frac{1+\alpha}{2}$ and the second receives $\frac{1-\alpha}{2}$.

Intuitively, we demonstrate that the above ratio is optimal when working against an adversary who can easily flip the input instance of Figure 2b for the two classes if the algorithm biases its decision making too heavily. We then show that the $\beta$ ratio is the best possible strategy against the possible adversarial input instances.

*Proof of Lemma B.8.* Our algorithm must ensure that neither class is envious of the other (more than the aforementioned threshold, $\beta$). Otherwise, the adversary has an instance that one class is envious of the other one and contradicts the threshold. Here, we show that if it at any iteration the $\beta$-CEF guarantee holds, we can strategically divide the next item to maintain the necessary inequality.

For the given construction, we can always match items to Class 2 which implies that $V_2^*(Y(X_1)) = V_1(X_1)$. Therefore, in achieving a $\beta$-CEF we must match as much of each item to the first class as possible to achieve equitable treatment. In other words, we give the maximum possible proportion of each item to the first class in a way that it does not contradict with $\beta$-CEF.

We will proceed by induction argument on the input stream. For the base case, assume towards contradiction that the first arriving item is not allocated according to the specified distribution, i.e. Class 1 is not fully saturated and the item is not distributed with ratio $1 + \alpha$ to $1 - \alpha$ to the first and second classes respectively. Observe that we cannot give more than a $\frac{1+\alpha}{2}$ fraction of the arriving item to Class 1 without violating the $\beta$-CEF guarantee.

For each of the subsequent items, we must match up to a $\frac{1+\alpha}{2}$ fraction of the item or less if all agents become saturated in Class 1. In each instance, the remaining portion of the item is given to the second class. We proceed to prove that if the allocation was $\beta$-CEF after each item's division, then this property must persist. To this end, assume that the argument holds for $t - 1$ for some $t \geq 2$.

First, we examine the second classes' perspective. We claim that matching the arriving item at iteration $t > 1$ according to the prescribed ratio ensures $V_2(X_2^t) \geq \beta \cdot V_1(X_1^t)$. At round $t$, we must have that this inequality was true prior to iteration $t$ by the induction assumption and by matching item $o_t$ according to the distribution between the two classes. Assume towards contradiction that we do not match according to the defined ratios at iteration $t$ – by matching the arriving item with a portion higher than $\frac{1+\alpha}{2}$, an adversary can instead flip the input instance and force the algorithm to allocate a smaller portion of the item than is needed for the $\beta$-CEF guarantee. Thus, after allocating item $o_t$ we cannot have given more than $\frac{1+\alpha}{2}$ to the first class and we still have $V_2(X_2) \geq \beta \cdot V_1(X_1) = \beta \cdot V_2^*(Y(X_1))$, where the last equation follows from the fact that we can always match every item to Class 2 that was matched to Class 1.

We lastly claim that $V_1(X_1) \geq \beta \cdot V_1^*(Y(X_2))$. From the perspective of the first class, we must have that the $\beta$-CEF inequality holds after allocating the first item according to the defined ratio. Following the prior item's arrival, we demonstrated that the inequality held. From the construction, we further have that each item is connected to more agents compared to items arriving later. To be more precise, if item $o_1$ arrives before item $o_2$, $o_1$ is connected to a set of agents, let's say $S_1$, and $o_2$ is connected to a set of agents, let's say $S_2$. Then $S_1$ is a superset of $S_2$. As a result, when we divide the arriving item, we have possibly decreased the portion of the next items (which were connected to fewer agents in class 1) and instead increased the portion of the current item (which is connected to a superset of the agents the next items are connected to). Therefore, if previously we had $V_1(X_1) \geq \beta \cdot V_2(X_2)$, then the inequality persists.

$\square$

Combining the results of Lemma B.7 and Lemma B.8, we have conditions under which we maintain a $\beta$-CEF approximation. It remains to analyze the maximum such $\beta$ value which satisfies these conditions.

*Proof of Theorem 1.3.* We begin by bounding the size of the matching to each class.

First, we claim that Class 2 will be saturated and its valuation will be $n$. Since we have $2n$ items and our algorithm is non-wasteful, we must match items in each step until the classes are satisfied. Further note that Class 2 will *always* have an available matching by construction. Now since Class 1 can only match at most $n$ items, we must have that $V_2(X_2) = n$.

We next bound the final step $i$ after which no arriving items will be matched to Class 1 as a result of our equal distribution within the class (as proven in Lemma B.7). By synthesis of the two above lemmas, we have that Class 1 receives a $\frac{1+\alpha}{2} \cdot \frac{1}{n}$ portion of the first two items. Furthermore, the $i$-th pair of items will contribute $\frac{1+\alpha}{2} \cdot \frac{1}{n-i+1}$ to the size of Class 1's matching.Therefore, it is sufficient

to find the first value of $i$ such that

$$\frac{1+\alpha}{2} \sum_{0 \leq k < i} \frac{1}{n-k} = \frac{1+\alpha}{2} (H_n - H_{n-i}) \geq \frac{1}{2},$$

where $H_n$ is the $n$-th Harmonic number. By the Euler–Maclaurin formula [3], we have

$$H_n = \ln n + \gamma + \frac{1}{2n} - \epsilon_n, \quad 0 \leq \epsilon_n \leq \frac{1}{8n^2}$$

where $\gamma$ is the Euler-Mascheroni constant. We now define $\epsilon = \frac{1}{2n} - \epsilon_n - \left( \frac{1}{2(n-i)} - \epsilon_{n-i} \right)$ and rewrite the desired expression as

$$1 = (1+\alpha)(H_n - H_{n-i})$$
$$= (1+\alpha)(\ln n - \ln (n - i) + \epsilon).$$

This further implies

$$\ln n - \ln(n - i) = \frac{1}{1+\alpha} - \epsilon. \tag{1}$$

which is equivalent to the following

$$\frac{i}{n} = 1 - e^{-\frac{1}{1+\alpha} + \epsilon}. \tag{2}$$

Since $i$ is the last item that is partially matched to Class 1, Lemma B.8 guarantees that this class receives a $1 + \alpha$ fraction of the item. Therefore, we must have that the matching to Class 1 is

$$\beta \leq \frac{i(1+\alpha)}{n},$$

where the factor of $n$ comes from the bound on Class 2's matching and the inequality from the algorithm's approximation guarantee. Expanding on this ineqality using Equation (2) implies

$$\beta \leq \frac{i(1+\alpha)}{n}$$
$$= (1+\alpha) \left( 1 - e^{-\frac{1}{1+\alpha} + \epsilon} \right)$$
$$= \frac{2}{1+\beta} \left( 1 - e^{-\frac{(1+\beta)}{2} + \epsilon} \right)$$
$$= \left( 1 - e^{\frac{-(1+\beta)}{2}} \cdot e^{\epsilon} \right) \frac{2}{1+\beta},$$

It remains show the limiting behavior of this bound on our approximation ratio as $n$ increases.

*Claim* B.9. $\lim_{n \to \infty} \epsilon = 0$

*Proof.* By definition of $\epsilon$ and the bound of $\epsilon_n < \frac{1}{8n^2}$, we can compute

$$|\epsilon| = \left| \frac{1}{2n} + \frac{1}{2(n-i)} + (\epsilon_{n-i} - \epsilon_n) \right|$$
$$\leq \frac{1}{2n} + \frac{1}{2(n-i)} + \frac{1}{8(n-i)^2}$$
$$\leq \frac{1}{2} + \frac{1}{2} + \frac{1}{8}$$
$$\leq 2.$$

Now by invoking Equation (2) we can expand the definition of $\epsilon$ as

$$|\epsilon| \leq \frac{1}{2n} + \frac{1}{2n} \cdot e^{\frac{1}{1+\alpha} - \epsilon} + \frac{1}{8} \left( \frac{1}{n} \cdot e^{\frac{1}{1+\alpha} - \epsilon} \right)^2$$

We lastly compute the limit:

$$\lim_{n\to\infty} |\epsilon| \leq \lim_{n\to\infty} \left( \frac{1}{2n} + \frac{1}{2n} \cdot e^{\frac{1}{1+\alpha} - \epsilon} + \frac{1}{8} \left( \frac{1}{n} e^{\frac{1}{1+\alpha} - \epsilon} \right)^2 \right)$$

$$\leq \lim_{n\to\infty} \left( \frac{1}{2n} + \frac{1}{2n} \cdot e^{\frac{1}{1+\alpha} - 2} + \frac{1}{8} \left( \frac{1}{n} e^{\frac{1}{1+\alpha} - 2} \right)^2 \right)$$

$$= 0$$

completing the claim. □

Using the result of Claim B.9, we have that $\lim_{n\to\infty} e^{\epsilon} = 1$ and thus by taking $n \to \infty$ we have

$$\beta \leq \left( 1 - e^{\frac{-(1+\beta)}{2}} \right) \frac{2}{1+\beta}$$

By direct computation, we obtain that $\beta \leq 0.677$. Therefore, we cannot achieve $\beta$-CEF for $\beta > 0.677$. □

## B.4 Proofs from Section 5

*Proof of Theorem 5.1.* Suppose that an algorithm, $\mathcal{A}$, is $\alpha$-CEF for some $\alpha \in (0,1)$. Let $p$ and $q$ be coprime integers such that $|\alpha - p/q| < \epsilon$ for some small $\epsilon > 0$. Assume we have $k-1$ classes with $q$ agents, as well as a $k$-th class comprised of $q(k-1)$ agents. An adversary constructs an input stream wherein the items arrive in two phases. In the first phase, $p(k-1) + q$ items arrive, each of which has edges to *every* agent in the graph. The second phase consists of $k-1$ groups arriving sequentially, where the $i$-th such group is comprised of $q$ items with edges to all the agents in class $i$. Let $c_i(t)$ be a random variable that indicates the number of items allocated to class $i$ at the end of round $t$ and let $\tau = p(k-1) + q$. We first prove the following claim for the given instance.

*Claim B.10.* For any non-wasteful $\alpha$-CEF algorithm and $\tau = p(k-1) + 1$, we must have that $\mathbb{E}\left[ \sum_{i=1}^{k-1} c_i(\tau) \right] \geq p(k-1)$.

*Proof.* Assume towards contradiction that $\mathbb{E}\left[ \sum_{i=1}^{k-1} c_i(\tau) \right] < p(k-1)$. Since the algorithm is assumed to be non-wasteful, this implies that the $k$-th class must have $\mathbb{E}\left[ c_k(\tau) \right] \geq q$ to account for all the items arrived thus far. By the pigeonhole principle, our assumption further implies that there exists some $i \in [k-1]$ such $\mathbb{E}\left[ c_i(\tau) \right] < p$. Thus, we must have that

$$V_i(X) = \mathbb{E}\left[ c_i(\tau) \right] < p < q \leq \mathbb{E}\left[ c_k(\tau) \right] = V_i^*(Y_k(X))$$

contradicting our assumption on the $\alpha$-CEF guarantee. □

We now proceed to analyze the class matching size in the second phase of item arrivals. Of the $q$ arriving items specific to some class $i$, exactly $c_i(\tau)$ must remain unmatched since all feasible agents will already be saturated. This implies that the utilitarian social welfare at the end of the second phase is

$$\text{USW}(X) = \left( \sum_{i=1}^{k-1} c_i(\tau) + c_k(\tau) \right) + \sum_{i=1}^{k-1}(q - c_i(\tau)) = \sum_{i=1}^{k} c_i(\tau) + \sum_{i=1}^{k-1} q - \sum_{i=1}^{k-1} c_i(\tau)$$

Now, using the fact that $p(k-1) + q$ items arrive in the first phrase where all can be matched to any agents, we further have that

$$\text{USW}(X) = p(k-1) + q + \sum_{i=1}^{k-1} q - \sum_{i=1}^{k-1} c_i(\tau) = p(k-1) + qk - \sum_{i=1}^{k-1} c_i(\tau)$$

where the remaining equalities are mere algebra manipulation on the summations. Now, as a result of Claim B.10 we have that in expectation:

$$\mathbb{E}\left[\text{USW}(X)\right] = \mathbb{E}\left[p(k-1) + qk - \sum_{i=1}^{k-1} c_i(\tau)\right] \leq qk.$$

Lastly, observe that the optimal offline solution for this adversarial input instance would instead allocate all items in the first phase to class $k$, and the remaining $q$ items specific to each class to their corresponding class, giving a USW value of $p(k-1) + q + q(k-1)$. Thus, the competitive ratio for the USW objective is given by

$$\frac{qk}{qk + p(k-1)} = \frac{1}{1 + \alpha(k-1)/k}$$

which tends towards a lower bound of $\frac{1}{1+\alpha}$ as $k$ increases. Thus, we have the result of the theorem. $\quad\square$

