# OpenReview forum: "Fairness and Efficiency in Online Class Matching"
_NeurIPS.cc/2024/Conference — NeurIPS 2024 poster_

### Official Review · Reviewer_yvhP · 2024-06-22

**Soundness:** 3
**Presentation:** 3
**Contribution:** 3
**Rating:** 7
**Confidence:** 3

**Summary:**

This paper studies the online bipartite matching problem where the agents (offline nodes) are partitioned into multiple classes.
 Upon the arrival of an item (online node), it needs to be immediately matched to a free agent or discarded, and the goal is to optimize fairness among different classes and efficiency simultaneously.

The main contribution of this paper is a simple randomized non-wasteful algorithm that simultaneously guarantees $1/2$-approximate class envy-freeness (CEF), $1/2$-approximate class proportionality (CPROP), and $1/2$-approximate utilitarian social welfare (USW). This is the first randomized non-wasteful algorithm that achieves class fairness guarantees. Furthermore, this paper complements this positive result by showing that no randomized non-wasteful algorithm can admit $\alpha$-CEF with $\alpha > 0.761$. When items are divisible, this paper also shows that no deterministic algorithm can achieve better than $0.67$-CEF.

Finally, this paper studies the notion called the price of fairness, which characterizes the trade-off between fairness and efficiency. This paper shows that any randomized algorithm that achieves $\alpha$-CEF cannot achieve strictly better than $\frac{1}{1 + \alpha}$-USW.

**Strengths:**

This paper studies a well-motivated and interesting problem. Also, this paper is well-written and well-structured.

All the results shown in this paper are non-trivial, and I believe that the community will be interested in them. In particular, I found the algorithmic result quite elegant and natural, for which the paper also provides interesting and technical analysis.

The price of fairness notion seems to be a good complement to prior results, and it would be exciting to know the trade-off curve between possibly conflicting objectives.

**Weaknesses:**

While the price of fairness has been extensively considered by the fair division literature (e.g., [1]), this paper doesn't provide any pointers to this thread of research.

Minor:
- Line 151: Should $\textbf{y}$ be a bundle instead of an allocation?
- Line 165: Typo: $V_i(Y_j^*(X))$.
- Line 197: "is is".
- In the paragraph starting from Line 238, there is no description for relating $Y_j$ to $A_i$.
- Line 255: "admits".

[1] The Price of Fairness for Indivisible Goods. Xiaohui Bei, Xinhang Lu, Pasin Manurangsi, Warut Suksompong.

**Questions:**

For Line 249-250, when constructing the vector for item $o$, does the vector depend on the history including the previously arrived items and the actions made by the algorithm so far? In both cases, it's not obvious to me why the probability in Line 251 holds. Can the authors further explain?

**Limitations:**

Certain references are missing.

---

> ### Author Rebuttal · Authors · 2024-08-06
>
> We thank the reviewer for their kind words with respect to our results and their presentation. We also appreciate the noted typos which have since been corrected.
>
> For the noted weakness on comparisons to other price of fairness (PoF) results: the first set of results on the PoF refers back to Bertsimas et al. and Caragiannis et al. Bertsimas et al. analyze the upper bound on the utility (egalitarian social welfare) loss incurred by fairness notion such as proportional fairness and max-min fairness for divisible goods. In particular, a main takeaway of their results is that for a small number of players, PoF stays relatively small, e.g., for two players PoF for proportional fairness is at most 8.6% and 11.1% for max-min fairness. Caragiannis et al. further obtain more extensive results from a similar vein, in particular for notions of proportionality, envy-freeness, and equitability, for allocation of divisible and indivisible goods and chores. As pointed out by Bei et al., their significant limitation for indivisible setting is that their guarantee does not hold for every problem instance as the results are not satisfied as a worst-case analysis. Bei et al. tackle such limitations by investigating PoF from worst-case scenario, under various notions including Nash social welfare, envy-free up to one good, balancedness, and egalitarian social welfare. We note that all of these results do not directly apply to our setting since our notion of class envy-freeness is not equivalent to any of the properties above.
>
> To your final question, the x_{a,o} are independent of rounding outcomes in the prior steps, hence the inequality holds for each agent being selected at least once with the given probability bound.
>
> *References:*
>
> Bertsimas et al.: The Price of Fairness, D. Bertsimas, V. Farias, N. Trichakis, OR’11
>
> Caragiannis et al.: The Efficiency of Fair Division, I. Caragiannis, C. Kaklamanis, P. Kanellopoulos, M. Kyropoulou, TCS’12
>
> Bei et al.: The Price of Fairness for Indivisible Goods, X. Bei, X. Lu, P. Manurangsi, W. Suksompong, TCS’21

---

> > ### Comment · Reviewer_yvhP · 2024-08-13
> >
> > Thanks for your response! I will keep my score positive.

---

### Official Review · Reviewer_h2C8 · 2024-07-10

**Soundness:** 3
**Presentation:** 2
**Contribution:** 2
**Rating:** 4
**Confidence:** 3

**Summary:**

The paper studies the online bipartite matching problem from a fairness perspective. One set of the bipartition (agents) are known a priori, and the other set (items) arrive online, wherein edges to that item are revealed and the algorithm must (possibly) select an edge to include in the matching before the next item is arrives. It is well known that the best competitive ratio for the basic problem is (1-1/e).

In this work, the agents are also partitioned into disjoint “classes” (e.g., demographic groups, interest groups, etc.). The goal is to compute an approximately fair and efficient matching under the following notions:
- Efficiency:
    - Competitive ratio on the size of the matching computed — the traditional objective, here referred to as the utilitarian social welfare or USW in keeping with the agent interpretation.
    - Non-wastefulness — there should not be any agent-item pairs with both the agent and item unsaturated but an unselected edge between them. In other words the matching should be maximal. Implies 1/2-approx on the previous USW objective via the typical maximal matching argument.
- Fairness:
    - Class envy-freeness (CEF) — In the end, no class of agents should be able to redistribute the total allocation of items to another class amongst themselves in a way that would lead to a larger matching than that assigned to the group. Approximations are multiplicative on these total matching sizes.
    - Class Proportional Fairness (CPROP) — A given class considers the best case over all possible (even divisible) global matchings, of the minimum over classes of the optimistic total matching size they might expect by redistributing the items assigned to that other class amongst themselves. The matching satisfies CPROP if every class gets at least this total matching size. Approximation is again multiplicative on the matching size.

The paper provides four primary results:
1. For the positive result, a randomized algorithm is studied that assigns each new item to a class uniformly at random, among those with an agent liking that item, and then to an agent within that class liking the item uniformly at random. This algorithm is non-wasteful by definition and thus 1/2-USW by corollary. The authors further show that the algorithm achieves a 1/2 approximation to CEF and CPROP.
2. The paper also provides impossibility of approximation results. In the same indivisible matching setting as above, hey show that no non-wasteful algorithm can achieve better than ~0.761-approximate CEF.
3. Furthermore, in the divisible matching setting (where items can be fractionally allocated), no non-wasteful algorithm can achieve better than ~0.67-approximate CEF.
4. Finally, an inverse proportionality is shown to exist between approximation of CEF and approximation on the USW objective.

**Strengths:**

The model, while shared with prior work, is a nice extension to the existing online fair division literature particularly in the subadditivity of class valuations due to the matching constraints, and the paper provides a nice analysis the expected class envy under a randomized algorithm that may be of independent technical interest. The paper also provides tighter upper (impossibility) bounds for approximating CEF non-wastefully and (in the price of fairness result) while approximating USW.

The writing is generally clear. I particularly appreciate that proof sketches of at least the constructions for the counterexamples are provided in the main body along with diagrams to aid the reader. The paper also does a good job of surveying relevant related work in context.

**Weaknesses:**

The positive algorithmic contribution seems very limited, as the only algorithm described (Algorithm 1) is essentially to allocate at random (random non-wasteful class, then a random non-wasteful agent within that class).

The relationship to the immediate prior work “Class Fairness in Online Matching” and the novel contribution of the current work are not clearly described. For example, lines 63-65 introduce the current results by saying “…we provide the first non-wasteful algorithm that simultaneously obtains approximate class fairness guarantees in expectation...” although the prior work cited above also defines non-wasteful algorithms that obtain approximate class fairness guarantees; the difference being that those are ex-post guarantees from deterministic algorithms rather than merely in expectation. More specifically, that prior work developed a deterministic algorithm for indivisible items for the same problem that achieves a deterministic/ex-post (rather than merely in expectation) guarantee of non-wastefulness, 1/2-USW, 1/2 MMS (maximin share), and 1/2-CEF1.

Motivation is discussed in terms of prior work asking about randomized algorithms, but that was presumably motivated by a desire to achieve stronger guarantees rather than simply to introduce randomization for its own sake, and it is not clear that the new results under random allocation are significantly stronger. While the novel analysis technique is nice and the impossibility results are helpful, the significance and impact of the positive algorithmic result seems unclear overall.

Minor comment: Several of the references should be updated to refer to the peer-reviewed and published versions of papers rather than the archival pre-prints. For example, the immediate prior work the current paper follows up on, “Class Fairness in Online Matching” is referenced from arXiv rather than the published 2023 AAAI paper.

**Questions:**

Could the authors clarify the sense in which the current algorithmic results for the indivisible case are a substantial improvement over those obtained in the prior work Class Fairness in Online Matching? In particular, why is (1/2)-CEF and (1/2)-CPROP in expectation to be preferred to (1/2)-CEF1 and (1/2)-MMS ex post, and in what sense(s) is the Random algorithm superior or an improvement over the Match-And-Shift algorithm?

**Limitations:**

No concern

---

> ### Author Rebuttal · Authors · 2024-08-06
>
> We thank the reviewer for their comprehensive feedback and kind words regarding the problem setting and approximate results (both upper and lower bound).
>
> For the first noted weakness on the algorithmic contribution, we refer the reviewer to our general comments made to Reviewer gpQ6 on the approximate guarantee and nontriviality of our algorithm.
>
> For the inquiry regarding the preferences for (1/2)-CEF and (1/2)-CPROP in expectation over (1/2)-CEF1 and (1/2)-MMS ex post, as well as the comparison between the Random algorithm and the Match-And-Shift algorithm, we emphasize that the prior work was unable to provide any approximate guarantees for the CEF objective. Moreover, (1/2)-CEF is a stronger guarantee than their (1/2)-CEF1 by definition, as it ensures a greater degree of equity between groups while maintaining the same approximate fairness factor. It is also known in the fair division literature that EF1 allocations are guaranteed to exist and are computable in polynomial time whereas EF is not guaranteed to exist. For proportionality, (1/2)-CPROP is fundamentally stronger than (1/2)-MMS. CPROP considers the maximum minimal satisfaction achievable over all possible (including divisible) matchings, whereas MMS is limited to indivisible matchings. This distinction means that CPROP offers a more robust and flexible framework for ensuring fairness.
>
> The simplicity of the Random algorithm and its non-trivial yet strong O(1) expected guarantees makes it highly practical and promising. The inherent flexibility of randomized algorithms allows for a degree of leeway in the allocation process, accommodating minor errors or misestimations without significant degradation of the overall fairness or efficiency of the system. This flexibility as compared to deterministic algorithms is crucial in real-world applications where exact outcomes may be unpredictable or where slight deviations from the ideal are acceptable for the sake of greater overall efficiency or user satisfaction.
>
> Furthermore, the Random algorithm is nearly optimal at the class level (see our impossibility construction), simple to implement while ensuring the above noted improved and more practical expected guarantees. This makes the Random algorithm superior in environments where conditions are dynamic or partially unknown at the time of decision-making. We believe these methods offer significant advantages for complex allocation tasks, particularly when equity and adaptability are paramount.
>
> Lastly, we thank the reviewer for noting the potentially outdated bibtex entries. We have gone through and rectified the references which needed updates. We hope the reviewer will consider increasing their score if the above sufficiently addresses the concerns.

---

> > ### Comment · Reviewer_h2C8 · 2024-08-10
> >
> > I acknowledge that I have read the author rebuttal.
> >
> > Thank you for the disccusion around my concerns and questions. I appreciate and agree that CPROP is stronger than MMS and CEF is stronger than CEF1, but wish to reemphasize my point that, for example, (1/2)-CEF in expectation is incomparable with (1/2)-CEF1 in the worst case, and that, in my opinion, it is somewhat misleading to characterize the results as the first non-wasteful class fairness guarantees when Class Fairness in Online Matching, AAAI 2023 also gives a non-wasteful algorithm with a (different and incomparable) class fairness guarantee.
> >
> > I appreciate that the randomized algorithm may be simpler to implement in a complex environment, and that is a reasonable advantage.

---

> ### Author Response · Authors · 2024-08-11
>
> We appreciate the reviewer for being very responsive.
> We agree that ex-ante guarantee and ex-post guarantee are usually not directly comparable, however, it is explicitly mentioned as one of the main open problems in Hosseini et al. 2023:
> 1. Any deterministic algorithm for "divisible" items with CEF better than 1-1/e and NW
> 2. Any deterministic algorithm for "divisible" items with any reasonable CEF, NW, and USW better than 1/2.
> 3. Any randomized algorithm for "indivisible" items with any reasonable CEF, NW and USW.
>
> In particular, their randomized algorithm (nor deterministic algo) does not have any CEF guarantee, but only has 0.593-PROP, 1/2-USW and NW.
> In addition, to provide such a missing hole in the randomized result (of no CEF guarantee), they further analyze another wasteful algorithm with CEF guarantee, but it is necessarily bound to be wasteful, which we believe is the reason that they pose open problem 3 explicitly.
> Hence, we believe our result in this context explicitly resolves open problem 3 with further guarantee on CPROP.
>
> We hope this meets the reviewer's expectations and would appreciate your understanding as well as a possible increase in the score in par with other reviewers.

---

### Official Review · Reviewer_CL5G · 2024-07-12

**Soundness:** 3
**Presentation:** 3
**Contribution:** 3
**Rating:** 6
**Confidence:** 3

**Summary:**

The paper addresses the online bipartite matching problem with a focus on class fairness, proposing the first randomized non-wasteful algorithm that balances class envy-freeness, class proportionality, and utilitarian social welfare. It introduces the concept of the "price of fairness," highlighting the trade-off between fairness and optimal matching, where increased fairness results in decreased utilitarian social welfare.

**Strengths:**

-The work resolves a long-standing conjecture by developing a non-wasteful randomized algorithm that achieves non-trivial fairness guarantees in online matching problems, complementing existing bounds on class envy-freeness (CEF), class proportionality (CPROP), and utilitarian social welfare (USW).
- It introduces the concept of the "price of fairness", showing the inverse proportionality between fairness and optimality, and presents an impossibility result demonstrating the trade-off between achieving CEF and maximizing USW.

**Weaknesses:**

- The motivated applications are not clear in the current context.
- No experimental results are provided to show the actual performance of the proposed algorithm.

**Questions:**

none

---

> ### Author Rebuttal · Authors · 2024-08-06
>
> For an expansion on motivating examples of this problem setting: the study of online matching under fairness constraints is motivated by the challenges posed by the advent of Internet economics and new marketplaces, which demand solutions that are both transparent and fair, as highlighted in Moulin’s “Fair Division in the Internet Age”. These applications, where items must be matched to agents immediately and irrevocably as they arrive, include allocating advertisement slots [Mehta et al., 2007], assigning packets in switch routing [Azar and Richter, 2005], distributing food donations [Lee et al., 2019], and matching riders to drivers in ridesharing platforms [Banerjee and Johari, 2019]. Much of this work ignores the notion of fair matching. For example, a food bank that is distributing food must allocate food as it arrives due to perishability, and it’s important to ensure that these resources are sent to locations in such a manner that serves all communities equitably.
>
> With respect to the lack of experimental validation, we emphasize that this work is intended to be largely theoretical. Capturing this problem precisely with an experimental procedure raises several design questions and would be an interesting direction of future research. Specifically, since our analysis is worst-case it would be interesting to see how our algorithm performs in practice.
>
> We hope the reviewer will consider increasing their score if the above sufficiently addresses the concerns!

---

### Official Review · Reviewer_gpQ6 · 2024-07-12

**Soundness:** 3
**Presentation:** 3
**Contribution:** 3
**Rating:** 6
**Confidence:** 3

**Summary:**

This paper studies the online bipartite matching problem with class fairness guarantee. In this problem, the offline vertices are divided into $k$ classes, and the challenge is to match each online vertex to an unsaturated offline vertex while providing guarantees on various fairness metrics (e.g., class envy-freeness (CEF), class proportionality (CPROP)) and efficiency metrics (e.g., utilitarian
11 social welfare (USW)). Previous work [16] has designed deterministic non-wasteful algorithms to achieve tight guarantees for these metrics.

This paper introduces the first non-wasteful randomized algorithm that simultaneously guarantees 1/2-CEF, 1/2-CPROP, and 1/2-USW. It also presents more general upper bound results (for both deterministic and randomized algorithms) for CEF in both indivisible and divisible settings. Additionally, this paper formalizes the price of fairness in the fair matching setting, providing an upper bound on efficiency (USW) for a given CEF guarantee.

**Strengths:**

- This paper represents the first attempt to study non-wasteful randomized algorithms with a CEF guarantee. The analysis of the studied RANDOM algorithm is non-trivial. It also provides general upper bounds (including for randomized algorithms) for CEF, which supplements the existing results.

- The hard instances used to prove the upper bound are interesting and can be useful for future research.

- The topic of online matching with class fairness guarantees is important and timely in the field of algorithmic fairness, although I feel a bit concerned regarding whether this theoretical problem is sufficiently motivated for the NeurIPS community.

**Weaknesses:**

- The authors claim to have resolved the open problem raised by ref [16]: Can a randomized algorithm for matching indivisible items achieve any reasonable CEF approximation together with either non-wastefulness or a USW approximation? However, I am not fully convinced that a "reasonable" CEF means 1/2, as a non-wasteful deterministic algorithm that provides such guarantees already exists (Theorem 1 in ref [16]). Although the analysis for the current randomized algorithm is non-trivial, the significance of a 1/2 CEF is unclear.

- Typos, e.g., in line 165 page 4, $V_i(Y_j^*(X)) \to V_i^*(Y_j(X))$

**Questions:**

- Can you explain the significance of the 1/2-CEF1 guarantee for non-wastefulness? For example, are other natural randomized algorithms challenging to achieve $\alpha$-CEF with $\alpha > 1/2$ while ensuring non-wasteful or USW?

**Limitations:**

Yes

---

> ### Author Rebuttal · Authors · 2024-08-06
>
> We thank the reviewer for their careful reading of our work, highlighting the novelty of our analysis and impossibility constructions, as well as constructive feedback. We also appreciate the identification of a typo which has since been rectified. We here address the noted weaknesses and related question raised.
>
> For the open problem raised by [16] on non-wasteful randomized algorithms: it is important to highlight that the 1/2-CEF guarantee provided by our algorithm represents a significant advancement in the field, effectively resolving the open question from prior work. This result is the first of its kind to ensure a nontrivial guarantee of fairness in expectation under the stringent non-wasteful constraint. The noted nontriviality of analyzing the natural “random” algorithm to achieve this level of fairness without wasteful allocations underscores the novelty and utility of our approach. We also emphasize that the original Hosseini paper was only able to achieve an approximate 1-1/e guarantee with a randomized algorithm when allowing for wastefulness. By restricting ourselves to non-wasteful and fair allocations, our task becomes considerably more challenging. For a more extensive comparison of our result against that of [16], we refer the reviewer to our remarks to reviewer h2C8 which delineates how the fairness guarantees differ (and are improved).
>
> Moreover, as discussed in our upper bound construction for a randomized algorithm, we highlight that an optimal algorithm must implement a randomization procedure at the class level. An algorithm that does not incorporate this can be exploited by an adversary. Thus, although our algorithm seems natural, it is strongly motivated by this consideration. The nontrivial analysis, which relies on doubling techniques to properly bound the expected guarantees, will likely be useful in all future works that analyze algorithms abiding by these constraints.
>
> We acknowledge that surpassing the 1/2-CEF barrier remains a formidable challenge, one that we pose as an open problem to the research community. We hope that our work will inspire future research on this problem. Furthermore, we hope the reviewer will consider increasing their score if the above sufficiently addresses the concerns!

---

> > ### Comment · Reviewer_gpQ6 · 2024-08-10
> > **Response to rebuttal**
> >
> > Thank you for your reply. I agree that the randomized algorithm achieves a stronger guarantee in expectation (1/2-CEF and 1/2-CPROP) compared to the deterministic algorithm (1/2-CEF1 and 1/2-MMS) in [16]. However, I still believe that the goal of introducing randomization should be primarily to achieve a CEF guarantee greater than 1/2.
> >
> > After reading your rebuttal and reconsidering the paper's potential influence, I think that proposing an initial attempt at addressing this challenging problem and bringing the open question to the field are good contributions. I have therefore increased my score to 6.

---

### Decision · Program_Chairs · 2024-09-25

**Decision:**

Accept (poster)

**Comment:**

Reviewers found the topic is interesting and the approaches (mostly theoretical analysis) are solid and can be non-trivial. The disagreement is mostly on subjective perceptions of the contributions: Some reviewers are not satisfied with the positive result, while the other reviewers are less concerned. During the discussion, a negative reviewer still felt that the contributions are limited and exaggerated, while a positive reviewer agreed that the algorithm is not exciting but liked its simplicity. After all, the paper seems to make a non-trivial step forward. We hope the authors find the reviews helpful. Thanks for submitting to NeurIPS!